# Genome Analysis of *Phytophthora nicotianae* JM01 Provides Insights into Its Pathogenicity Mechanisms

**DOI:** 10.3390/plants10081620

**Published:** 2021-08-06

**Authors:** Xiao-Long Yuan, Cheng-Sheng Zhang, Fan-Yu Kong, Zhong-Feng Zhang, Feng-Long Wang

**Affiliations:** 1Tobacco Research Institute of Chinese Academy of Agricultural Sciences, Qingdao 266101, China; yuanxiaolong@caas.cn (X.-L.Y.); kongfanyu@caas.cn (F.-Y.K.); zhangzhongfeng@caas.cn (Z.-F.Z.); 2Special Crops Research Center of Chinese Academy of Agricultural Sciences, Qingdao 266101, China

**Keywords:** *Phytophthora nicotianae*, genome, divergence time, pathogenic mechanisms, comparative genomics

## Abstract

*Phytophthora nicotianae* is a widely distributed plant pathogen that can cause serious disease and cause significant economic losses to various crops, including tomatoes, tobacco, onions, and strawberries. To understand its pathogenic mechanisms and explore strategies for controlling diseases caused by this pathogen, we sequenced and analyzed the whole genome of *Ph. nicotianae* JM01. The *Ph. nicotianae* JM01 genome was assembled using a combination of approaches including shotgun sequencing, single-molecule sequencing, and the Hi-C technique. The assembled *Ph. nicotianae* JM01 genome is about 95.32 Mb, with contig and scaffold N50 54.23 kb and 113.15 kb, respectively. The average GC content of the whole-genome is about 49.02%, encoding 23,275 genes. In addition, we identified 19.15% of interspersed elements and 0.95% of tandem elements in the whole genome. A genome-wide phylogenetic tree indicated that *Phytophthora* diverged from *Pythium* approximately 156.32 Ma. Meanwhile, we found that 252 and 285 gene families showed expansion and contraction in *Phytophthora* when compared to gene families in *Pythium*. To determine the pathogenic mechanisms *Ph. nicotianae* JM01, we analyzed a suite of proteins involved in plant–pathogen interactions. The results revealed that gene duplication contributed to the expansion of Cell Wall Degrading Enzymes (CWDEs) such as glycoside hydrolases, and effectors such as Arg-Xaa-Leu-Arg (RXLR) effectors. In addition, transient expression was performed on *Nicotiana benthamiana* by infiltrating with *Agrobacterium tumefaciens* cells containing a cysteine-rich (SCR) protein. The results indicated that SCR can cause symptoms of hypersensitive response. Moreover, we also conducted comparative genome analysis among four *Ph. nicotianae* genomes. The completion of the *Ph. nicotianae* JM01 genome can not only help us understand its genomic characteristics, but also help us discover genes involved in infection and then help us understand its pathogenic mechanisms.

## 1. Introduction

Oomycetes are a kind of ubiquitous filamentous microorganisms that can cause various destructive diseases and cause economic losses of agricultural crops, ornamental plants, and natural plant populations [1,2,3]. Horizontal and endosymbiotic gene transfer events, gene fusions and duplications shaped and diversified the genomes of species in oomycetes [4,5]. Among which, species that belong to the genera *Phytophthora*, *Pythium*, and *Albugo* are the most diverse and destructive pathogens [1]. Therefore, many studies have been conducted to understand their virulence mechanisms, diversities and evolutionary positions, to identify targets for drugs, and crop protection chemicals to control disease. The virulence mechanisms and evolution of oomycetes have mainly been examined through transcripts and whole-genome sequences [6,7,8,9]. For example, the divergence times of oomycetes have been estimated under distinct molecular clock models, indicating the oomycete diverged about 430–400 MYA. Additionally, the divergence time of two major lineages Saprolegniales and Peronosporales in oomycetes were placed in the early Mesozoic (~225–190 MYA) [6]. Meanwhile, transcriptomics and comparative transcriptomic analyses have been performed among species in oomycetes to identify their transcriptional remodeling when the host was exposed to infection and chemical environments. For example. RNA-seq analysis of *Phytophthora capsici* identified stage-specific genes including effector families and metabolic pathways, revealing that these proteins are important in the infection process [10]. In addition, transcriptome analysis of three life stages of *Ph. litchii* predicted 490 pathogenicity-related genes [11]. Similar, RNA-seq of *Aphanomyces euteiches* showed that the adaptability to plant hosts was related to the expression of specialized secretions [12]. Importantly, studies of oomycetes have been advanced by the completion of their genome sequences. For example, the genome sequence of *Saprolegnia parasitica* revealed that some pathogenesis-related genes in this species originated from lateral gene transfer (LGT), thereby indicating the evolutionary adaptation to its hosts [13]. The completion of the genome of *Pythium ultimum* helped us understand a new pathogenic mechanism, which characterized by lacking crucial effectors such as Arg-Xaa-Leu-Arg (RXLR) effectors and Crinkler genes and gaining a novel YxSL family [8]. *Py. insidiosum* can cause infectious disease for both humans and animals, and its virulence and pathogenicity were previously examined at the genome level [14]. Moreover, a 38.7 Mb *Py. destruens* genome was generated by whole-genome shotgun sequencing technique [15]. *Phytophthora* is a notorious oomycete genus that contains more than 100 species and can cause damage to a range of economically important agricultural and horticultural plants [2]. Analysis of the genome of *Ph. sojae* revealed genes related to plant infections such as transporters, protein toxins and proteinase inhibitors [16]. *Ph. infestans* is a model oomycete that causes destructive disease of potatoes, resulting in huge economic losses. In 2009, the *Ph. infestans* genome revealed the notable rapid expansion and evolution of effector genes [6]. The 64 Mb genome of the pepper and cucurbits pathogen *Ph. capsica* was published in 2012 [17], while its genome size was corrected to 100.5 Mb by the long-read sequencing method in 2021 [18]. Additionally, the genomes of cacao black pod rot pathogens *Ph. megakarya* and *Ph. palmivora* were also assembled in 2017 [19]. In 2018, Vetukuri et al. reported the genome sequences of *Ph. plurivora* and *Ph. colocasiae*, and analyzed their pathogenicity-related genes, such as RXLR, Crinkle, and hypersensitive response effectors [20]. In addition, Yang et al. generated a high-quality genome assembly of *Ph. vignae* strain PSY2020 using Oxford Nanopore Technologies (ONT) sequencing and identified 924 candidate RXLR effectors [21]. In addition, comparative genomic analysis among species in oomycetes showed that genes related to secreted effectors and CWDZs exhibited expansion in *Phytophthora*, while genes encoding proteolytic degradation and signal transduction were increased in *Pythium*, which provided important insights into the evolutionary history of oomycetes and their mechanisms of pathogenesis [22].

*Phytophthora nicotianae* is widely distributed in soil and irrigation water, and is one of the most important pathogens in the Peronosporales, which can survive in field soil for several years [23]. For tobacco, it can cause black shank disease in different tissues, and cause root rot, leaf wilting, black stems and even death, which greatly reduces tobacco production and causes significant economic losses [24]. Records of the private company indicated that the occurs of *Ph. nicotianae* and can cause losses of about 65% or more in Colombia. Additionally, the average annual economic loss caused by *Ph. nicotianae* in China is more than one hundred million yuan [25]. Previous studies have identified four physiological races of this pathogen (0, 1, 2 and 3) in major tobacco cultivation areas worldwide [26]. Among these races, the most common strain, race 0 has been the predominant race in the flue-cured area of North Carolina since 1931 [27]. Race 1 is pathogenic on *N. plumbaginifolia*, but its fitness was not equivalent to race 0, and it can be induced under intense selection pressure of continuous breeding of race 0 resistant varieties [28]. Races 2 and 3 have been reported in South Africa and Connecticut, respectively [29,30]. Previous studies of *Ph. nicotianae* mainly focused on morphological and physiological identification, pathology, evolution, and prevention measures [31,32,33]. However, this disease is difficult to control via chemical means. Thus, the breeding of pathogen-resistant cultivated lines currently presented is the major control strategy to control this pathogen. Genome information can yield tremendous information that helps us understand its evolutionary position and pathogenic mechanisms. Currently, there are already three assembled genomes of *Ph. nicotianae* in NCBI database (*Ph. nicotianae* race 0, race 1 and BL) [26]. The genomes *Ph. nicotianae* race 0 and race 1 were published in 2016 by a combination of PacBio single-molecular real-time sequencing and second-generation sequencing methods, with a genome size of 81 Mb and 71 Mb, respectively. Genome analysis showed that the *Ph. nicotianae* race 0 and race 1 contain expanded ATP-binding cassette transporter gene family when compared with those in *Ph. infestans*. This result suggests that the ATP-binding cassette transporters plays important roles in the adaptive evolution of *Ph. nicotianae* to the host [26]. In addition, Liu et al. (2016). found that the difference in RXLR effector gene numbers between race 0 and race 1 may be related with their pathogenicities in cultivated tobacco [26]. The genome size of *Ph. nicotianae* BL was about 106.75 Mb. However, no systematic studies have been performed for their divergence time and virulence mechanisms.

Therefore, we isolated *Ph. nicotianae* JM01 from tobacco and sequenced its whole genome and analyzed its secreted proteins and effectors to investigate its pathogenic mechanisms. We estimated the divergence time of *Ph. nicotianae* JM01 based on molecular time markers and compared its genome with some closely related oomycetes. These results help us understand the pathogenic mechanism of *Ph. nicotianae* JM01 and provide a theoretical basis for future disease prevention and control.

## 2. Results

### 2.1. Genome Sequencing, Assembly and Validation

Sequencing reads that were generated from Illumina and PacBio platforms were used for assembling the *Ph. nicotianae* JM01 genome after quality control. A total of ~22.14 Gb and ~5.32 Gb of clean data were obtained from shotgun sequencing platform (21.20 Gb for 170 bp, 10.68 Gb for 500 bp, 3.50 Gb for 2000 bp and 1.42 Gb for 6000 bp libraries) and single-molecule sequencing platform, respectively. For the Hi-C library, we produced 28.37 Gb clean data. First, we obtained a draft genome of *Ph. nicotianae* JM01, with a total length of 59.05 Mb, with an N50 contig and scaffold length of 54.01 kb and 100.23 kb, respectively. After assembling the contigs, we used mate-pair sequencing data and PacBio sequences to generate scaffolds. Combined with the Hi-C data, we obtained a genome of approximately 95.68 Mb in size, consisting of 1446 scaffolds. The contig and scaffold N50 are 54.23 kb and 176.88 kb, respectively (Table 1). To assess the quality of the assembled genome of *Ph. nicotianae* JM01, we analyzed the completeness of the gene sets using BUSCO. The results showed that 92.37% of the eukaryotic single-copy genes were detected in the genome of *Ph. nicotianae* JM01 (Appendix A).

### 2.2. Repeat Elements

Repeat elements constituted 20.10% of the *Ph. nicotianae* JM01 genome, with 19.15% interspersed repeats and 0.95% tandem repeat sequences, respectively. Among the identified interspersed repeats, we found that 9461 long terminal repeated LTRs (3,660,470 bp) covered 7.62% of the whole genome, among which 7785 (6.38%) belonged to the Gypsy superfamily and 1179 (0.83%) belonged to the Copia superfamily. Additionally, we identified 214 (0.16%) LTRs that belonged to the Ngaro superfamily. DNA elements, including MuLE-MuDR, TcMar-Tc1, and PiggyBac, comprised 3.52% of the genome (Table 2).

After repeat masking, de novo prediction, homology searches, and RNA-based methods were used to predict gene structures of *Ph. nicotianae* JM01. *De novo* prediction using AUGUSTUS 3.0.2 predicted 17,325 gene models for *Ph. nicotianae* JM01 [34]. For homologous annotation, *Ph. nicotianae* JM01 genome scaffolds were used as queries to search for a database containing protein sequences from seven *Phytophthora* species (*Ph. capsici*, *Ph. ramorum*, *Ph. infestans*, *Ph. parasitica*, *Ph. cinnamomic*, *Ph. nicotianae* race 0 and *Ph. sojae*) and accordingly identified 45,276 homologous protein sequences. We also mapped the cDNA and ESTs of *Ph. nicotianae* to the genome using BLAST and then assembled using PASA [6]. We totally predicted 15,652 gene models. Finally, EVM was used to integrate the above results and obtain a gene dataset comprising 23,275 protein-encoding genes. Most of these genes contained few introns, with an average of 1.59 introns per gene, and were relatively short (~157.36 bp). The coding exons and coding sequences of the *Ph. nicotianae* JM01 genome showed average lengths of 592.77 bp and 1789.27 bp, respectively.

To further determine the functions of the protein-coding genes in *Ph. nicotianae* JM01, these protein sequences were aligned against several public databases. We accordingly identified 23,130 genes that had at least one hit in the NR database, which have high homology with species in the genus *Phytophthora* (99.38%). The top five species for which we obtained the best BLAST hits for *Ph. nicotianae* JM01 are *Ph. parasitica*, *Ph. infestans*, *Ph. sojae*, *Ph. halstedii,* and *Albugo laibachii* in NR database (Appendix A). Based on the results of GO analyses, we successfully assigned a total of 6592 protein-coding genes to GO glossaries with 23,551 functional terms (Appendix A). KEGG analysis revealed that 2367 genes have KO annotations (Appendix A), and a total of 337 KEGG metabolic pathways were annotated in the *Ph. nicotianae* JM01 genome (Appendix A). Furthermore, we found that 6550 genes in the *Ph. nicotianae* JM01 genome could be assigned to 1704 CDD superfamilies. Moreover, Pfam annotation showed that a total of 9789 genes in the *Ph. nicotianae* JM01 genome were associated with 30,927 domains.

### 2.3. Gene Family Expansion and Contraction

To compare gene families and contents among the different species, we performed orthology analysis. The results indicated the expansion and contraction of 574 and 125 gene families, respectively, in the *Ph. nicotianae* JM01 genome when compared with that of *Hyaloperonospora arabidopsidis*. In addition, we identified 252 and 285 gene families in *Phytophthora* that showed expansion and contraction when compared with those in *Pythium*. In *Phytophthora*, we found that among the gene families identified, 136 (containing 728 genes), 13 (127 genes), 24 (216 genes), 15 (17 genes), 48 (1959 genes), 39 (812 genes), 14 (14 genes), 36 (2514 genes), 31 (222 genes) and 32 (1595 genes) were uniquely expanded in *Ph. cinnamomi*, *Ph. capsici*, *Ph. nicotianae* JM01, *Ph. nicotianae* race 1, *Ph. nicotianae* BL, *Ph. infestans*, *Ph. nicotianae* race 0, *Ph. parasitica*, *Ph. ramorum,* and *Ph. sojae*, respectively. In *Ph. nicotianae* JM01, we identified 417 and 994 gene families that were expanded and contracted, respectively.

### 2.4. Phylogenetic Relationship and Divergence Time Analyses

To elucidate the evolutionary position of *Ph. nicotianae* JM01, we performed phylogenetic relationships and divergence time analysis based on 46 single-copy genes. The constructed phylogenetic tree showed that *T. thermophila*, used as an outgroup, was located at the basal position of the whole tree (Figure 1). We identified four distinct lineages in the tree: the brown algae, diatoms, Saprolegniales and Peronosporales. A clade united brown algae and diatoms together, which emerged as a sister group to the clade containing Saprolegniales and Peronosporales. Among the Peronosporales, we found that *Ph. nicotianae* initially clustered with *Ph. parasitica*. A further clade placed the genera *Phytophthora, Hyaloperonospora*, *Pythium*, and *Albugo* together. In addition, we found that *Pythium vexans* was located close to a clade containing *Phytophthora* and *Hyaloperonospora*, and then clustered with the genus *Pythium*. Furthermore, our relaxed molecular clock analysis indicated the *T. thermophila* initially appeared at 705.53 Ma (95% highest posterior density (HPD) = 637.62–789.30 Ma). Similarly, we determined that brown algae, diatoms, and Saprolegniales initially appeared at 532.18, 220.91, and 102.12 Ma (95% HPD = 466.62–572.21, 197.03–245.15 and 91.93–112.61 Ma), respectively. We also estimated that the ancestor of Peronosporales originated in the Carboniferous, approximately 348.26 (327.73–385.68) Ma. Among the genera in the Peronosporales, the divergence events were involved in the emergence of *Hyaloperonospora* (87.00(78.66–92.77) Ma), *Pythium* (230.38 (212.34–255.02) Ma) and *Albugo* (293.13 (269.48–318.74) Ma), and we found that species of *Phytophthora* shared a common ancestor at approximately ~58.21 (53.91–63.12) Ma. Additionally, the results showed that all the four strains of *Ph. nicotianae* diverged from each other approximately at ~2.96 (2.38–3.65) Ma.

### 2.5. Analysis of Pathogenicity Mechanisms

The first stage for *Phytophthora* to infect the host is to destruct their cell wall polysaccharide components using carbohydrate-active enzymes (CAZys), which also promotes their utilization of nutrients in the host cell. Our analysis indicated that glycosyl transferases (GTs), glycoside hydrolases (GHs), polysaccharide lyases (PLs), carbohydrate esterases (CEs), auxiliary activities (AA) and the carbohydrate-binding module (CBL) have been expanded in the *Ph. nicotianae* JM01 genome (Figure 2A). Based on the CAZy database, we identified 44 unique carbohydrate enzymes in infected hosts that are unique to the *Ph. nicotianae* JM01 genome. Moreover, AA3, AA4, AA6, AA7, CBM1, CBM13, CBM47, CE8, CH1, CH10, CH109, CH12, CH131, CH140, CH16, CH17, CH28, CH3, CH30, CH43, CH16, CH17, CH28, CH3, CH30, CH43, CH5, CH72, GT1, GT31, PL1 and PL3 underwent significant expansion in *Ph. nicotianae* JM01 genome. Analysis of the phylogenetic relationships of GH genes revealed that the expansion of these genes duplicated in the *Ph. nicotianae* JM01 genome (Figure 2B).

Elicitins can induce a hypersensitive response in some species due to its structures. We observed an expansion of these proteins in *Phytophthora* genomes compared with those in *Pythium* genomes. We identified 42, 60, 48, 32, 51, 54, 57, 54, 54 and 67 elicitins in *Ph. capsici*, *Ph. cinnamomi*, *Ph. infestans*, *Ph. nicotianae* race 0, *Ph. nicotianae* JM01, *Ph. nicotianae* BL, *Ph. nicotianae* race 1, *Ph. ramorum* and *Ph. sojae*, respectively. Exhibiting cellulose-binding, the elicitor of plant defense responses, and lectin-like activities (CBELs), interferons (INFs), oligopeptide elicitors (OPELs) and Peptides (PEPs) are important elicitors in pathogenic organisms. Our results indicated the presence of 16 CBELs, 36 INFs, 16 OPELs, and 21 PEPs in the *Ph. nicotianae* JM01 genome (Appendix A), which is comparable to the composition in other species.

Many plant pathogens can produce an abundance of secreted proteins which can infect their hosts’ immunity. To explore the pathogenic mechanisms of *Ph. nicotianae* JM01, we selected secreted proteins and effectors from known related species as databases. Finally, we identified 2445 predicted secreted proteins in the *Ph. nicotianae* JM01 genome, which is a much larger number than those found in other *Phytophthora* species to date. Among the secreted proteins in the *Ph. nicotianae* JM01 genome, we identified 211 RXLR effectors, which is broadly comparable to the numbers in other *Phytophthora* species, including *Ph. capsica* (357), *Ph. cinnamomic* (173), *Ph. infestans* (323), *Ph. nicotianae* race 0 (133), *Ph. nicotianae* race 1 (163), *Ph. nicotianae* BL (156), *Ph. parasitica* (230), *Ph. ramorum* (143), and *Ph. sojae* (264) and considerably more than that in the *H. arabidopsidis* genome (29). Similar to other *Phytophthora* species, the RXLR effectors in *Ph. nicotianae* JM01 contain amino-terminal cell entry domains with RXLR and dEER motifs (Figure 3A–C). For investigating the functions of these genes, we performed phylogenetic analyses to establish their relationships. The results showed that gene duplication played an important role in RXLR genes expansion (Appendix A).

Crinkler (CRN) represents a type of cytoplasmic effector gene, which encodes a secreted protein containing a conserved amino-terminal LFLAK domain. We identified 24 CRN genes in *Ph. nicotianae* JM01 genome. In contrast to RXLR effectors, the structure of CRN genes showed high conservation in most pathogenic oomycete genomes that have been published to date (Figure 3D). In addition, a total of 97, 37, 196, 22, 24, 26, 28, 86, 79 and 19 CRN genes have been identified in *Ph. cinnamomi*, *Ph. capsici*, *Ph. infestans*, *Ph. nicotianae* race 0, *Ph. nicotianae* JM01, *Ph. nicotianae* BL, *Ph. nicotianae* race 1, *Ph. parasitica*, *Ph. ramorum,* and *Ph. sojae*, respectively. We aligned the amino-terminal regions of all 24 *Ph. nicotianae* JM01 CRN proteins and found that these shared conserved LxLYLAR/K and WL motifs (Appendix A). PcF (*Phytophthora cactorum*-*Fragaria* protein), a cysteine-rich (SCR) protein, was also found only in *Ph. parasitica*, *Ph. infestans*, *Ph. cinnamomi*, *Ph. ramorum* and *Ph. sojae.* In the *Ph. nicotianae* JM01 genome, we identified a SCR protein PnF1, a homolog of PcF. To investigate the function of PnF1, transient expression was performed on *N. benthamiana* by infiltrating with *A. tumefaciens* cells containing PnF1. On day 4 post-treatment, the negative control leaves still remain healthy, whereas those infiltrated with PnF1 showed symptoms of hypersensitive response (Figure 4A). The results showed that the areas were infiltrated with BAX and INF1 showed hypersensitive reaction, while the areas injected with GFP and MgCl_2_ did not show any symptoms of hypersensitive response (Figure 4B). Cutinase is another cysteine-rich (SCR) protein, which is defined as extracellular serine esterase that can break the ester bonds of cutin. Among *Phytophthora* species, a total of 15 cutinases were found in *Ph. sojae*, only two of which lacked a signal peptide. Interestingly, among the six cutinases in *Ph. capsica,* only one contained a signal peptide. Both *Ph. infestans* and *Ph. ramorum* have four cutinases, all of which contain a signal peptide. Generally, we found that cutinases are highly conserved in the genomes of *Phytophthora* species. In addition to these effectors, we searched for other secreted proteins in the *Ph. nicotianae* JM01 genome, including ATR13, HLE and TOXA. However, we did not find genes coding for these proteins.

Transcription factors can bind to DNA and regulate gene expression by either promoting or suppressing transcription. It has been demonstrated that transcription factors play important roles in pathogenicity by mediating oxidative stress response during plant defense responses and virulence. We identified 19 proteins as potential basic region—leucine zipper (bZIP) proteins (Figure 2A). To further examine the characteristics of these bZIP domains, these sequences were compared with the conventional bZIP domains. As shown in Figure 2A, some of the conserved region comparisons showed differences between the *Ph. nicotianae* JM01 bZIP domains and conventional bZIP domains, especially in DNA recognition regions. The candidate bZIPs were divided into the following six classes: N-R, C-R, R-R, V-R, N-K and others. In addition, we also analyzed the bZIP proteins in other species, and accordingly found that the *Ph. nicotianae* JM01 genome contains a larger number of bZIP proteins than those of the other assessed species. Myb factor is also a type of transcription factor that plays an important role in the process of pathogen infection. In total, we identified 64 Myb in the *Ph. nicotianae* JM01 genome (Figure 2A), much larger than the numbers found in any other *Phytophthora* genome sequenced to date.

### 2.6. Comparative Genomes of Four Ph. nicotianae Genomes

The gene contents of *Ph. nicotianae* JM01 were compared with those of *Ph. nicotianae* race 0, *Ph. nicotianae Ph. nicotianae* race 1, and *Ph. nicotianae* BL. A total of 664 core genes shared by these four strains (Figure 5). These genes were mainly annotated as integrase core domain, growth-arrest specific micro-tubule binding, fibrinogen alpha/beta chain, protein tyrosine kinase, lipopolysaccharide kinase, Viral A-type inclusion protein, etc. Except for the common shared genes, *Ph. nicotianae* JM01, *Ph. nicotianae* race 0, *Ph. nicotianae Ph. nicotianae* race 1, and *Ph. nicotianae* BL contained 680, 2732, 6612 and 991 specific genes, respectively.

To investigate the genomic structure in four *Ph. nicotianae* genomes, MUMmer software (Version 3.23) was used to analyze the genome synteny. The syntenic regions comparison showed that these genomes shared common syntenic regions. However, some structural rearrangement such as inversion and translocation still happened after the divergence of these strains (Figure 6). In detail, the *Ph. nicotianae* JM01 and *Ph. nicotianae* race 0, *Ph. nicotianae* JM01 and *Ph. nicotianae* race 1, *Ph. nicotianae* JM01 and *Ph. nicotianae* BL shared 12,922, 12,179 and 14,429 blocks, respectively. Between *Ph. nicotianae* JM01 and *Ph. nicotianae* race 0 (Appendix A), 6269 locations experienced inversion, and 12,663 locations experienced translocation. Additionally, 3706 inversions and 11,840 translocations were found between *Ph. nicotianae* JM01 and *Ph. nicotianae* race 1 (Appendix A). In addition, we found that 5182 locations experienced inversion, and 6015 locations experienced translocation between *Ph. nicotianae* JM01 and *Ph. nicotianae* BL (Appendix A).

## 3. Discussion

### 3.1. Genome Features

*Ph. nicotianae* is one of the most significant important plant pathogens [2]. *Phytophthora*-associated diseases are difficult to control through chemical means and other strategies. Moreover, three genomes of *Ph. nicotianae* (*Ph. nicotianae* race 0, race 1 and BL) were available in the NCBI database. Liu et al. (2016). analyzed the RXLR effector and found that the difference in RXLR effector gene numbers between race 0 and race 1 may be related with their pathogenicities in tobacco [26]. However, there were no systematic studies which evaluated the divergence time and virulence mechanisms of *Ph. nicotianae*. Therefore, we assembled the genome of *Ph. nicotianae* JM01 using multiple techniques to understand its pathogenicity mechanisms. The assembled genome of *Ph. nicotianae* JM01 is about 95.32 Mb in size, which is close to the previously published genomes of *Ph. sojae* and *H. arabidopsidis* [16,35], although smaller than that of *Ph. infestans*, with the assembled genome size of 240 Mb. Previous studies showed that the estimated genome size of *Ph. ramorum* is 65 Mb [3], which is smaller than that of *Ph. nicotianae* JM01 [16]. Furthermore, the genome size of *Ph. nicotianae* races 0, *Ph. nicotianae* race 1 and *Ph. nicotianae* BL was about 81 Mb, 71 Mb and 106.75 Mb., respectively [26]. The genome sizes of *Ph. nicotianae* BL and *Ph. nicotianae* JM01 were similar, while they were much larger than those in race 0 and race 1. The assembled genome had a GC content of ~49.02%, which is comparable to those in other species [16]. The genome sizes of species in *Phytophthora* ranged from 52.4 Mb (*Ph. lateralis*) to 236.0 Mb (*Ph. alni var. alni*) [36]. Transposable and repetitive elements can generate mutations and alter the genome size, which can markedly impact the genomic architecture and evolution of species. In this regard, our comparison of repeat elements among the *Phytophthora* species revealed that 21% of the *Ph. nicotianae* JM01 genome comprises repeat elements, which is comparable to that in *Ph. ramorum* (18%) and *Ph. sojae* (28%), while it is lower than that in *Ph. infestans* (74%) [36]. Comparison of the available genome sequences indicated that repeat elements contribute to genome size differences in *Phytophthora* species. We found that prediction methods and databases of repetitive sequences have a great influence on the identification of repetitive sequences. Usually, repetitive sequences are removed before gene prediction. Therefore, the removal of repetitive sequences determines the accuracy of gene prediction. When we predict repeating sequences, we used RepBase in 2005 to perform the repeat element prediction, which may also be a limitation, especially for the gene family comparison. Additionally, we found that the sizes of *Pythium* genomes range from 33.9 Mb to 44.7 Mb [8]. Another reason for the varied genome sizes may be the different sequence approaches. For example, the 64 Mb genome of the *Ph. capsica* was corrected to 100.5 Mb in length by the long-read sequencing method in 2021 [18]. Moreover, the genome completeness evaluated according to core eukaryotic genes showed similarities among those in the published oomycete genomes, representing the accuracy and adequate coverage of the whole genome and gene sets.

### 3.2. Phylogenetic and Divergence Time Analyses

Phylogenetic analyses among these species suggested that the genera *Phytophthora* and *Pythium* are sister groups, which is agreement with previous studies [6]. In this study, we constructed phylogenic time trees for the brown algae, diatoms, Saprolegniales and Peronosporales based on the currently available genomes. We estimated the divergence time of brown algae, diatoms, and Saprolegniales at 532.18, 220.91 and 102.12 Ma (95% HPD = 466.62–572.21, 197.03–245.15 and 91.93–112.61 Ma), respectively. We also estimated that the ancestor of Peronosporales originated in the Carboniferous, at approximately 348.26 (327.73–385.68) Ma, which is similar to a previously published estimate [37]. In addition, comparative genome analysis showed that these four species shared a total of 664 core genes. According to the annotation results, we found that the core genes are closely related to cellular components, molecular functions, polysaccharide biosynthetic process, etc. In addition, large numbers of long syntenic regions were identified among *Ph. nicotianae* JM01, *Ph. nicotianae* race 0, *Ph. nicotianae* race 1 and *Ph. nicotianae* BL, implied that they are conserved in genome synteny. The genome sizes varied among these four species. This may be caused by repeat regions or the assembly approaches. The possible reason is that these four strains are physiological races of *Ph. nicotianae*. However, what we should pay attention to is that there are still a lot of translocations, which may be caused by the inconsistency and inversion of the chromosomes. In addition, we noticed that different genome sizes can also lead to the comparison of collinearity modules. For example, there are fewer inversions and translocations between *Ph. nicotianae* BL and *Ph. nicotianae* JM01 than the other two species.

### 3.3. The Pathogenic Mechanisms of Ph. nicotianae

In plants, the cell wall represents an important barrier to avoid invasion of pathogens and is also an important site of host plant–pathogen interactions. Polysaccharides, such as cellulose, hemicellulose, pectin, along with proteins and lignin are the main content of plant cell walls [38]. Extracellular enzymes produced by the plant pathogenic fungi can degrade the components of the plant cell wall. This not only provides sources of nutrients for their growth, but also promotes their colonization and spreads in tissues. The important role of CWDEs in the pathogenicity of phytopathogenic fungi has attracted the attention of many researchers [8]. In the present study, we identified 44 genes coding carbohydrate enzymes detected in infected hosts unique to the *Ph. nicotianae* JM01 genome, among which some of the AAs, CBMs, CHs and GTs have undergone significant expansion. It has been reported that the *Phytophthora* species harbors the expansion gene datasets encoding secreted effectors and CWDZs, differed from *Pythium* species whose over-represented genes were mainly related to proteolytic degradation and signal transduction [22], which is further supported by the results of our study. In addition, our analysis of the phylogenetic relationships among GH genes indicated that the duplication of these genes may be attributed to their expansion in the *Ph. nicotianae* JM01genome.

It has been demonstrated that CBEL can reduce or eliminate host defenses in *N. benthamiana* as a cell wall glycoprotein [39]. Previous studies showed that 13, 15, 2 and 2 genes have been identified as CBELs in *Ph. sojae*, *Ph. ramorum*, *H. arabidopsidis*, *A. candida*, *respectively* [35,40]. We found 16 genes encoding CBELs in the *Ph. nicotianae* JM01 genome, which is more than those in other closely related species.

Many effectors have been identified in plant pathogens, and numerous gene-poor regions have harbored rapidly evolving pathogenicity effectors. In the present study, we found a significantly larger number of predicted secreted proteins in the genome of *Ph. nicotianae* JM01 than those in the genomes of *Ph. sojae* and *Ph. ramorum*. The CRN and RXLR are mostly populated genes in *Phytophthora* species [6]. RXLRs have been identified that are abundant in *Phytophthora* species (Jiang et al., 2008). Analysis of the genome sequences of several genomes showed that the *Phytophthora* genome harbors extraordinarily large RXLR families, which were supported by our results. In 2018, Vetukuri et al. (2018) identified pathogenicity-related genes such as RXLR, Crinkle, and Necrosis effectors in *Ph. plurivora* and *Ph. colocasiae* genome [20]. Additionally, 924 candidate RXLR effectors were identified in the *Ph. vignae* genome [21]. In addition, Liu et al. (2016) reported that the difference in RXLR effector gene numbers between *Ph. nicotianae* race 0 and race 1 may be related to their pathogenicities in cultivated tobacco [26]. In our study, we found that some RXLRs had NUDIX, WYL3, MtN3_slv, Acyl_trasnf and MaoC_dehydratas motifs in the *Ph. nicotianae* JM01 genome, while their functions remain to be clarified in pathopoiesis. Brett et al. (2010) proposed that these motifs could aid in the protein entry of these proteins into host cells [41]. Currently, little is known about the function of RXLRs in *Ph. nicotianae* JM01, as few such effectors have been studied in this species. PSE1, a member of the RXLR family of effectors in *Ph. parasitica*, accumulates during appressorium-mediated penetration of host roots and suppresses cell death triggered by both the elicitin cryptogein in *Ph. cryptogea* and AvrPto avirulence proteins in *Ph. syringae* [42]. These observations indicated that *Ph. parasitica* effectors are involved in disease pathogenesis in the pathosystem involving this fungus and *Arabidopsis thaliana* [42]. CRNs, which are other important and conserved secreted proteins in the oomycete lineage [6,12]. We found a small family of CRN genes in the *Ph. nicotianae* JM01 genome when compared with those in other *Phytophthora species* [8]. However, we found the other elicitors and other effectors showed an expansion in the *Ph. nicotianae* JM01 genome, although these genes are typically similar to those reported in all other species of *Phytophthora*. Combined with the results of phylogenetic analysis, we predicted the expanded pathogenesis-related genes were gained by gene duplication, thereby accelerating the evolutionary adaptation to their hosts. A previous study showed that some pathogenesis-related genes in *S. parasitica* were derived from lateral gene transfer [43]. Different pathogens regulate their methods of infecting host cells through different evolutionary pathways.

Transcription factors existed extensively in different kingdoms. Moreover, it has been found that key elements in transcription factor also tend to be conserved, although there are exceptions, some of which are involved in several spore stages important for pathogenicity and play significant roles in the development of sporangia [4,44]. Pibzp1, which belongs to a bZIP family protein, was previously characterized in *Ph. infestans* and shown a relationship with zoospore motility and infection for their host [45]. In both fungi and plants, bZIPs have been found to regulate responses to the different abiotic stresses like starvation, osmotic and reactive oxygen etc. [4,45]. PsMYB1, an R2R3-type Myb transcription factor from *Ph. sojae*, which are regulated by PsSAK1 and play an important role in lifecycle stages during the infection process [46]. The function of bZIPs performed regulatory effects on *Ph. sojae* via interacting with promoter elements [47]. Meanwhile, some promoter elements have been identified based on the whole genome of *Ph. infestans*, which are closely related to spore formation and infection. The transcription factor Mybal is considered to play an important role in the spore development of oomycetes. This inference is supported by the function of PsMYB1, which is required for sporangium differentiation and the release of zoospores. In *Ph. infestans*, the expression PiMyb2R3, an ortholog of PsMYB1, is reduced during sporulation. We accordingly believe that analyses of the bZIP and Myb transcription factors may improve the understanding of gene regulation in *Ph. nicotianae* JM01. We identified a total of 64 Myb factors in the *Ph. nicotianae* JM01 genome, which is a larger number than those have been found in any other *Phytophthora* genome examined to date. Based on our analyses, we speculated that gene duplication also played important roles in the pathogenesis of *Ph. nicotianae* JM01 and formed a predicted mechanism of *Ph. nicotianae* JM01 (Appendix A). The results revealed that gene duplication has contributed to the expansion of CWDEs such as glycoside hydrolases, and effectors such as classical RXLR effectors. Thus, we speculated that the CWDEs in *Ph. nicotiana* JM01 can degrade the content of polysaccharides of tobacco leaves. Subsequently, the effectors, elicitors enter the host and gradually infect the host.

## 4. Materials and Methods

### 4.1. Material Culture

*Ph. nicotianae* (JM01) was isolated from infected tobacco plants in Yunnan Province in 2016 and stored in the microbiology laboratory of the Institute of Tobacco Research Institute of Chinese Academy of Agricultural Sciences. The mycelium was cultivated on V8 medium at 25 °C. Before extracting DNA, the mycelium was collected and then washed with sterilized distilled water for the following use.

### 4.2. DNA Extraction and Quantification, Genome Sequencing

Genomic DNA was isolated using a standard phenol/chloroform protocol [48]. Breifly, 100 mg samples were quickly ground in liquid nitrogen and then added to a centrifuge tube containing 500 μL lysate. Then, the homogenous material was centrifuged at 12,000× *g* for 5 min. Next, the supernatant was transferred to another tube, and equal volume phenol-chloroformisoamyl alcohol (25:24:1) added. Subsequently, the mixture was slowly homogenized for 2 min and centrifuged at 12,000× *g* for 5 min. Then, the nucleic acids were precipitated by addition of 1000 uL of 75% ethanol twice. Nucleic acids were pelleted at 12,000× *g* for 10 min. The pellet was dried and was resuspended with 30 μL of ultrapure sterile water and stored at −80 °C. DNA concentration was evaluated using a NanoPhotometer spectrophotometer (IMPLEN, Westlake Village, CA, USA). Then, the high-quality DNA (260/230 ≥ 1.8, 260/280 ≥ 1.8) was used to construct two pair-end DNA libraries with an insert length of 170 bp and 500 bp, respectively, and two mate-pair libraries with an insert length of 2000 bp and 6000 bp, respectively. For pair-end libraries, 1 µg of the DNA was sheared using a Covaris system (Covaris, Inc. Woburn, MA, USA) using the 170 bp or 500 bp program, respectively. Then, the sheared DNA fragments were purified with AMPure XP beads, end-repaired, dA-tailed, and ligated to Illumina universal adapters. After adapter ligation, DNA fragments were further used for size selection. In addition, the sequencing index was added to the former products. For mate-pair libraries, 10 µg (for 2000 bp and 6000 bp size libraries) of genomic DNA was sheared to the desired size by Hydroshear (Digilab, Marlborough, MA, USA), then used for end repair. Fragment sizes were purified from 1% low melting agarose gel and circularized by blunt-end ligation. Similarly, purification, end-repaired, dA-tailed, and ligated to Illumina PE sequencing adapters, along with the sequencing index ligation were performed. Finally, the purified libraries were detected using Agilent Bioanalyzer 2100 (DNA 7500 kit) before sequencing. These libraries were sequenced on the Illumina Hiseq2000 platform (San Diego, CA, USA), generating 100-bp pair-end reads. Trimmomatic [49] was used to remove adapters and sequencing primers from the raw data. Meanwhile, FastQC was used for controlling the quality of the raw data [50].

For SMRT library construction and sequencing, the SMRTbell genomic library was constructed using 5 μg DNA using SMRTbell template kit 1.0 (Menlo Park, CA, USA) according to the manufacturer’s instructions and was loaded on the RSII platform (P6C4 chemistry) for sequencing. The raw reads were trimmed for low-quality sequences and the organelle sequences were removed by BLASR [51]. Additionally, chromatin was fixed with formaldehyde, and the fixed DNA was extracted and digested with *Dpn*II for a Hi-C library construction. Subsequently, sticky ends were biotinylated and proximity ligated to form chimeric junctions, which were enriched, and then physically sheared into sequences ranging from 300 to 500 bp. These chimeric fragments representing the original cross-linked long-distance physical interactions were then processed into paired-end sequencing libraries.

### 4.3. Genome Assembly

For genome assembly, SOAP denovo v2.04 was used to assemble the *Ph. nicotianae* genome with default parameters based on the clean data from two pair-end libraries (170 bp and 500 bp) [52]. Then, the clean reads from 2000 bp and 6000 bp mate-pair libraries, and reads from the Pacbio platform, were used to construct scaffolds using SSPACE [53] and BLASR [51], respectively. Subsequently, the gaps that were produced during the scaffolding construction were closed using PBJelly2 with default parameters [54]. Furthermore, the proximity-guided assembly was also performed based on the Hi-C data used for scaffolding. In detail, the paired-end reads were uniquely mapped to the assembled genome, which then scaffolded using a 3D de novo assembly (3D DNA) pipeline (https://github.com/theaidenlab/3d-dna (accessed on 5 June 2020)) with tuned parameters. The raw data have been deposited in the public database NCBI with project number PRJNA389504. The assembled genome has been uploaded as Dataset 1.

### 4.4. Repeat Element Prediction and Masking

The assembled genome was first scanned to find repetitive elements to avoid interfering with the results of gene prediction. RepeatMasker v4.0.6 [55] and RepeatModeler [56] were used to search for homologous repeat elements in the RepBase [57] and for de novo repeat element identification in the assembled scaffolds, respectively. In addition, summary data were collected for the repeat types, scores, motifs, and positions of simple sequence repeats (SSRs). Tandem repeat sequences in the genomes of *Ph. nicotianae* were identified using the Tandem Repeats Finder [58], and repeat elements were masked before conducting gene predictions using RepeatMasker v4.0.6 [55].

### 4.5. Gene Prediction

Gene prediction for the masked genome of *Ph. nicotianae* JM01 was performed using three approaches: (1) de novo gene prediction was performed using AUGUSTUS 3.0.2 [34]; (2) protein evidence was downloaded from the UniProt/Swiss-Prot protein database [59,60] and the predicted proteins of *Ph. capsici*, *Ph. ramorum*, *Ph. infestans*, *Ph. parasitica*, *Ph. cinnamomic, Ph. nicotianae* race 0 and *Ph. sojae* using exonerate [60,61] and (3) EST or cDNA sequences of *Ph. nicotianae* that were downloaded from NCBI were selected to predict gene structures using PASA [6]. When all the prediction results were integrated, we assigned weight scores to different data sets, and then integrated these three prediction results, and finally formed the gene sets of the *Ph. nicotiana* JM01 genome. The transcripts are the direct evidence of coding genes. Therefore, we used the existing EST or cDNA sequence in the database to predict the gene model, and the weight score is 8 points. For Augustus, the weight score is 10 points; the protein-coding region based on the closely related species is given a weighted score of 7 points. Finally, EvidenceModeler (2009) was used to integrate the comprehensive set of genes and gene prediction results [6]. After the prediction of the EVM results and corrections by PASA, the final protein-coding sequence was 23,275. Then, genome assembly and protein completeness were assessed using Benchmarking Universal Single-Copy Orthologs (BUSCO) (version 2.0), which contains 4584 conserved core genes in eukaryotes [62].

### 4.6. Gene Annotation

For gene annotation, the predicted genes were annotated using different databases (Nr, GO, Interpro, Pfam and KEGG). Nr BLAST was performed locally using BlastP with an e-value of e^−5^ [63], and then Gene Ontology (GO) and InterPro were annotated based on the results from Blast2GO [64]. Pfam protein analysis was performed locally using an e-value of e^−5^ [65]. KEGG annotation was analyzed using the KAAS web server (https://www.genome.jp/tools/kaas/ (accessed on 19 September 2020)) [66].

### 4.7. Gene Expansion and Contraction Analysis

To analyze gene families and gene contents variation in *Ph. nicotianae* JM01 and other closely related species (Appendix A), we performed gene expansion and contraction analysis using OrthoMCL v4.0 software with an e-value of e^−5^ and proteins in closely related genomes with a similarity greater than 50% were defined as orthologous proteins [67].

### 4.8. Phylogenetic and Divergence Time Analyses

The phylogenetic tree was constructed using OrthoMCL v4.0 software based on common single-copy genes [67]. For tree construction, *Tetrahymena thermophila* was selected as an outgroup species. MAFFT v7 was used to perform alignments [68]. Finally, protein datasets were used to establish phylogenetic relationships and estimate divergence time using BEAST v1.7.5 [69]. The time constraint was chosen according to the divergence time between *Thraustotheca clavata* and *Ph. infestans* (~363 Ma) [66].

### 4.9. Identification of the Pathogenic Effectors of Ph. nicotianae JM01

To characterize the pathogenic mechanisms of *Ph. nicotianae* JM01, secreted proteins were identified using SignalP V5.0 [70]. Sequences containing transmembrane domains and organelle-targeting signals were removed from the above predicted secreted proteins. The transmembrane domains of secreted proteins can also be predicted using TMHMM2.0 (TransMembrane prediction using Hidden Markov Models) [71]. Finally, the functions of these proteins were determined using WEGO 2.0 [72]. For subsequent analyses, we analyzed the cell wall-degrading enzymes, elicitors, and effectors of *P. nicotianae* and available related species. The carbohydrate enzymes (CAZy) in the genomes were predicted using the CAZy Analysis Toolkit with default parameters. For elicitors prediction, we downloaded the known elicitors (INF1s, OPELs, CBELs and PEP-13s), and aggregated them into multiple sequence alignments, and then conducted BLAST analyses. For effectors analysis, hidden Markov models of the SCR, RXLR, ATR13, Elicitin, HLE, PCF, and TOXA effectors were downloaded from Pfam database (http://pfam.xfam.org/ (accessed on 25 January 2021)) and Hmmer software (Hmmsearch, Hinxton, UK) [73] was used to identify candidate genes in the genome that matched this model. We also used the Hmmer package hmmbuild to construct a CRN.hmm model according to known Crinkler genes (CRNs) and then used the hmmsearch package to explore CRN domains in *Ph. nicotianae* JM01.

### 4.10. Analysis of Basic Region-Leucine Zipper (bZIP) and V-Myb Avian Myeloblastosis Viral Oncogene Homolog (Myb) Sequences

For analysis of bZIP and Myb sequences in *Ph. nicotianae* JM01 genome, tools in Hmmer software [73] were used to format HMM models. Subsequently, the sequence alignment was performed using MUSCLE (www.drive5.com/muscle (accessed on 21 February 2021)) [74]. Then, the sequence logos of bZIP and Myb domains were generated using Hmmer software [73].

### 4.11. Validation of Ph. nicotianae JM01 Effectors by Constructing Recombinant Agrobacterium Tumefaciens Binary PVX Vectors

Cysteine-rich (SCR) protein is a secreted heparin-binding protein, which is related to cell surface, the extracellular matrix, and biochemical characteristics. To determine the function of effectors, a SCR protein (PnF1) was amplified using the DNA isolated from *Ph. nicotianae* JM01. The PCR product was cloned into the PVX vector pGR107 (Tyler et al., 2006). Then, the accuracy of recombinant binary plasmid was sequenced by Sanger sequencing. After confirmation, the constructs were propagated in *Escherichia coli*, which were cultured in LB medium containing 12.5 mg/mL tetracycline and 50 mg/mL kanamycin. Then, the constructs were selected under the same concentration of tetracycline and kanamycin mentioned above. Subsequently, these constructs were introduced into *A. tumefaciens* strain GV3101 via electroporation [8]. Finally, the recombinant strains of *A. tumefaciens* were infiltrated into *N. benthamiana* leaf using a needle to detect the function of PnF1. Meanwhile, BCL2-Associated X (BAX) (usually used as a positive control for hypersensitive response) and INF1 (which is an extracellular protein secreted by the oomycete *Ph. infestans*), were injected to *N. benthamiana* leaf as a positive control and green fluorescent protein (GFP) and 10 mM MgCl_2_ were selected as a negative control.

### 4.12. Genome-Wide Multicollinearity Analysis and Gene Contents Analysis among Four Ph. nicotianae Strains

To infer the evolution of genomic structure in four *Ph. nicotianae* genomes, we used MUMmer software (Version 3.23) to scan the genomes of *Ph. nicotianae* JM 01, race 0, race 1 and BL to identify putative homologous chromosomal regions. Then, we used LASTZ (Version 1.03.54) to compare these regions, and confirmed the arrangement relationships, and found the translocation, inversion, and translocation + inversion regions. In addition, the core genes and specific genes were identified using cd-hit (Version 4.6.1) software. Additionally, the R package was used to draw a Venn diagram [75].

## 5. Conclusions

*Ph. nicotianae*, widely distributed in soil and irrigation water, is one of the most significant important pathogens in the Peronosporales. We isolated *Ph. nicotianae* JM01 from tobacco, and then sequenced its whole genome and analyzed its genome features, evolutionary position, and its possible pathogenic mechanisms in this study. Furthermore, we performed a comparative analysis among four *Ph. nicotianae* genomes. The length of the assembled genome of *Ph. nicotianae* JM01 is about 95.32 Mb. Additionally, we identified 19.15% interspersed elements and 0.95% tandem elements across the whole genome. In addition, a genome-wide phylogenetic tree indicated that *Phytophthora* diverged from *Pythium* by approximately 156.32 Ma. To determine its pathogenic mechanisms, we analyzed a suite of proteins that were involved in plant–pathogen interactions. In addition, transient expression was performed on *N. benthamiana* by infiltrating with *A. tumefaciens* cells containing a cysteine-rich (SCR) protein to detect the function of effector. The genome synteny results showed large amounts of long syntenic regions were shared among *Ph. nicotianae* JM01 and *Ph. nicotianae* race 0, *Ph. nicotianae* JM01 and *Ph. nicotianae* race 1, and *Ph. nicotianae* JM01 and *Ph. nicotianae* BL. Additionally, 680 core common genes shared in these strains. In summary, our analyses of the whole-genome sequence of *Ph. nicotianae* JM01 reinforce previous hypotheses regarding the pathogenic mechanisms of *Phytophthora* and will help us advance the current understanding of infected mechanisms of oomycete plant pathogens.

## Figures and Tables

**Figure 1 plants-10-01620-f001:**
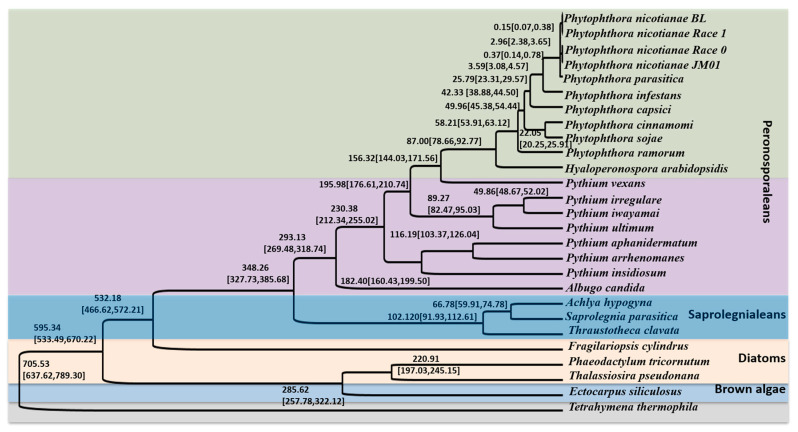
Phylogenetic relationships of *Ph. nicotianae* JM01and other species based on the dataset of common single copy genes. This phylogenetic tree was generated using Bayesian inference in BEAST v1.7.5. The time scale was set according to the divergence time between *T. clavata* and *Py. infestans* (~363 Ma). The divergence time of each species was marked above each branch with 95% confidence level. Different colors represent different groups.

**Figure 2 plants-10-01620-f002:**
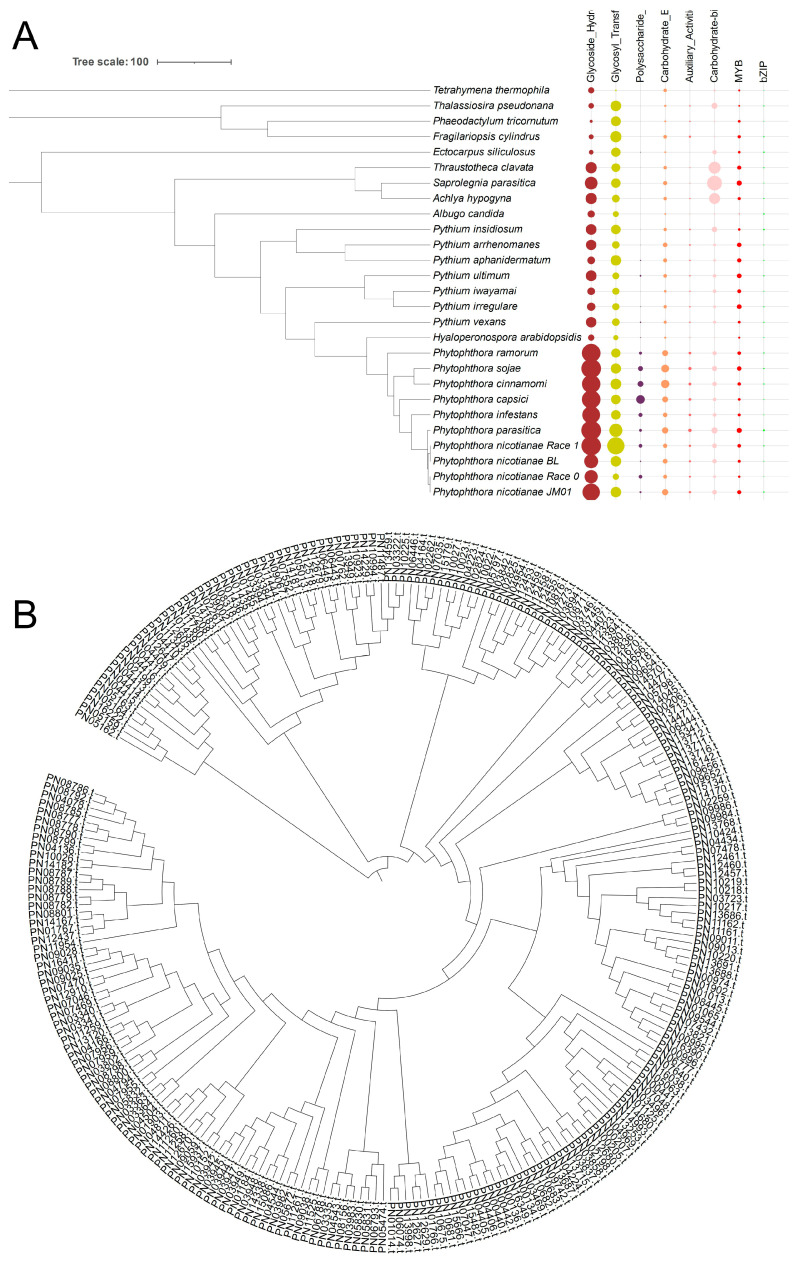
CAZyme annotation and transcription factors of *Ph. nicotianae* JM01 and other closely related genomes. (**A**) The CAZyme annotation and transcription factors results. The tree on the left was generated using Bayesian inference in BEAST v1.7.5. Different circles present glycoside hydrolases, glycosyl transferases, polysaccharide lyases, carbohydrate esterases, auxiliary activities, the carbohydrate-binding module, MYb and bZIP. The sizes of dots on the right present the gene numbers of different CAZyme genes and transcription factors. (**B**) Phylogenetic relationships of glycoside hydrolases (GH) genes in *Ph. nicotianae* JM01.

**Figure 3 plants-10-01620-f003:**
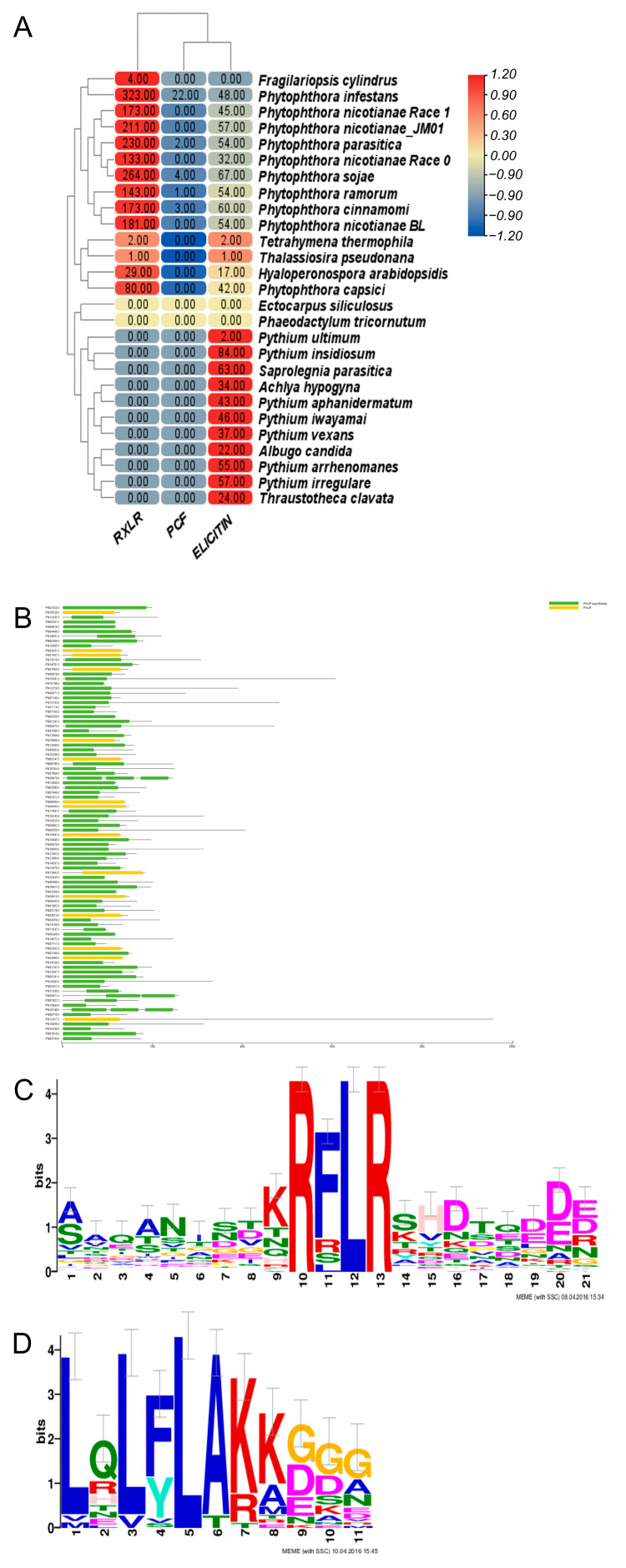
Prediction results of effectors in *Ph. nicotianae* JM01 genome and other species. (**A**) Results of effectors in the *Ph.nicotianae* JM01 genome and other species. A heatmap was used to present the gene numbers of RXLR, PCF and ELICITIN; (**B**) Genes structures of RXLR in *Ph.nicotianae* JM01. Conserved domains were drawn in NCBI; (**C**) Typical architecture of RXLR in *Ph. nicotianae* JM01; (**D**) Typical architecture of CRN in *Ph. nicotianae* JM01.

**Figure 4 plants-10-01620-f004:**
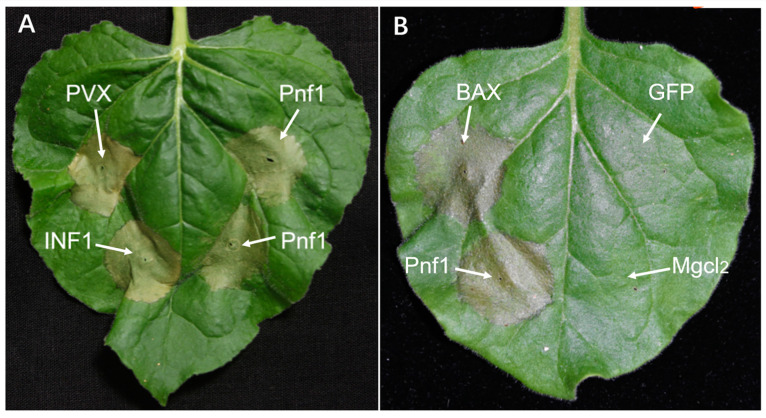
PnF1from *Ph. nicotianae* JM01 triggers cell death in *N. benthamiana* leaf. (**A**) The recombinant strains of *A. tumefaciens* containing PnF1 were infiltrated into *N. benthamiana* leaf using a needle to detect the function of PnF1. The upper left portion of the leaf in (**A**) is injected with PVX, and the lower left portion of the leaf in (**A**) is injected with INF1. The right portion of the leaf in (**A**) is injected with Pnf1. (**B**) The BCL2-Associated X (BAX) was selected as a positive control, and green fluorescent protein (GFP) and MgCl_2_ were selected as negative controls. The upper left portion of the leaf in (**B**) is injected with BAX, and the lower left portion of the leaf in (**B**) is injected with Pnf1. The upper right portion of the leaf in (**B**) is injected with GFP and the lower left portion of the leaf in (**B**) is injected with MgCl_2_.

**Figure 5 plants-10-01620-f005:**
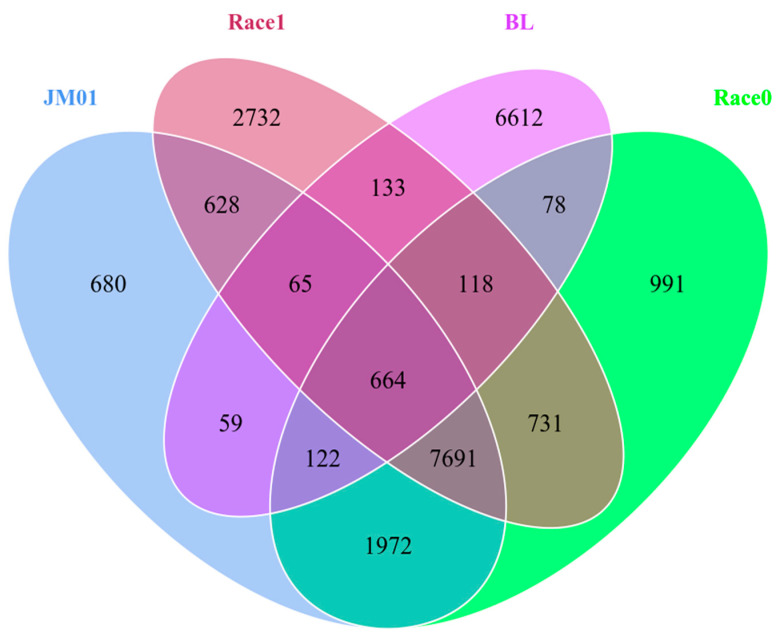
Venn diagram showing the core and pan genes among four *Ph. nicotianae* genomes. Names of each species were listed beside the diagram component. *Ph. nicotianae* JM01 (JM01); *Ph. nicotianae* race 0 (race 0); *Ph. nicotianae* race 1 (race 1); *Ph. nicotianae* BL (BL).

**Figure 6 plants-10-01620-f006:**
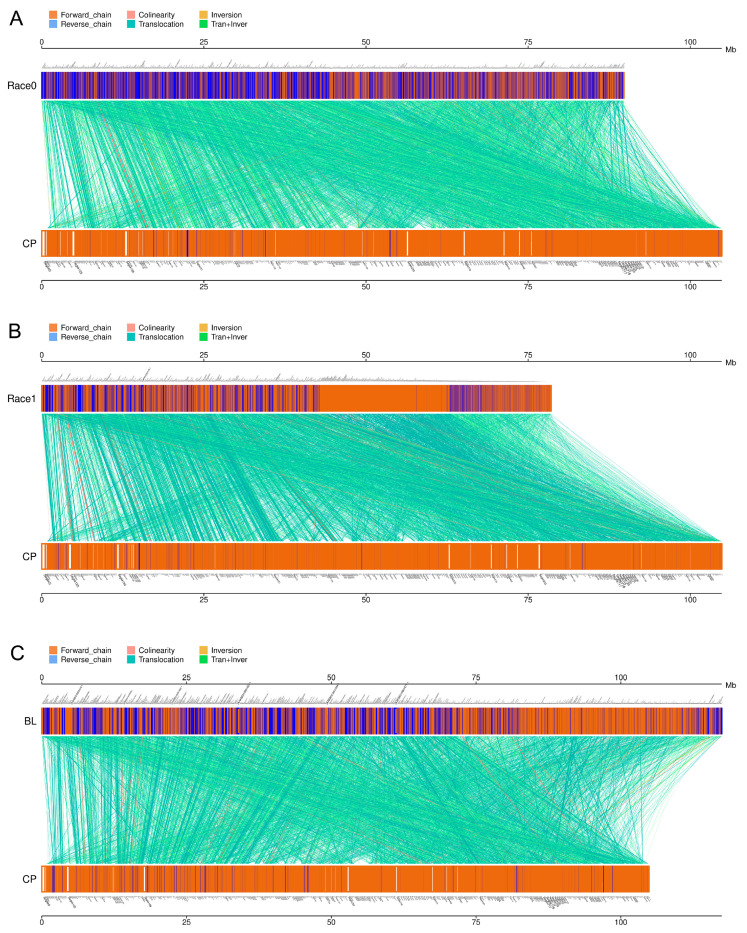
Syntenic regions comparison of four *Ph. nicotianae* genomes. The wide horizontal lines in red, blue, and green represent the assembled scaffolds in *Ph. nicotianae* JM01, *Ph. nicotianae* race 0, *Ph. nicotianae* race 1 and *Ph. nicotianae* BL. MUMmer software (Version 3.23) was used to analyze the genome synteny between any two genomes. (**A**) Collinear blocks between *Ph. nicotianae* JM01 and *Ph. nicotianae* race 0. (**B**) Collinear blocks between *Ph. nicotianae* JM01 and *Ph. nicotianae* race 1. (**C**) Collinear blocks between *Ph. nicotianae* JM01 and *Ph. nicotianae* BL.

**Table 1 plants-10-01620-t001:** Genomic features of *Phytophthora nicotianae*.

Categories	Scaffold	Contig
Total Number	2105	3868
Total length (bp)	95,318,700	58,288,754
Average Length (bp)	29,463.56	15,069.48
N50 Length (kb)	113.15	54.23
N90 Length (kb)	28.94	8.54
Maximum Length (bp)	540,321	383,516
Minimum Length (bp)	1002	200
GC content	49.02%	49.78%

**Table 2 plants-10-01620-t002:** Repeat elements in *Ph. nicotianae* genome.

Class	Type	Number of Elements	Length Occupied	Percentage of Sequence
Interspersed repeats	DNA elements			
	Crypton	535	320,797	0.34%
	Maverick	110	143,341	0.15%
	MuLE-MuDR	1243	587,998	0.62%
	MuLE-NOF	535	60,260	0.06%
	PIF-Harbinger	294	231,089	0.24%
	PiggyBac	450	257,994	0.27%
	Sola	275	91,598	0.10%
	TcMar-ISRm11	791	398,902	0.42%
	Merlin	198	88,846	0.09%
	TcMar-Pogo	201	54,855	0.06%
	TcMar-Ant1	272	138,354	0.15%
	TcMar-Tc1	807	449,528	0.47%
	TcMar-Tc2	469	215,152	0.23%
	TcMar-Stowawa	197	113,580	0.12%
	hAT-Ac	324	136,041	0.14%
	hAT-Tag1	176	68,793	0.07%
	DNA	543	303,342	0.32%
	LTR ^#^			
	LTR	283	238,869	0.25%
	Copia	1179	787,427	0.83%
	Gypsy	7785	6,079,720	6.38%
	Ngaro	214	148,614	0.16%
	RC ^$^			
	Helitron	654	444,976	0.47%
	Unknown	20279	6,888,937	7.23%
Tandem repeats	Microsatellite	717	52,963	0.06%
	Minisatellite	7437	598,915	0.63%
	Satellite	576	255,356	0.27%
Total		46544	19,156,247	20.10%

^#^ long terminal repeats; ^$^ rolling circle.

## Data Availability

The data presented in this study are available in this article and the Appendix A.

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
