# Peer review of "Genome Analysis of Phytophthora nicotianae JM01 Provides Insights into Its Pathogenicity Mechanisms"

_plants, 2021, doi:10.3390/plants10081620_

Round 1
Reviewer 1 Report
Very interesting work, well structured and accurate, of high scientific quality and soundness. I suggest to publish it.
Author Response
Comments and Suggestions for Authors
Very interesting work, well structured and accurate, of high scientific quality and soundness. I suggest to publish it.
Response:Thanks for your comments.
Reviewer 2 Report
Although the study is not extremely novel and other similar studies are present in the literature, in my opinion it was well conducted (experimental design).
I have some comments/ suggestions that they should be answered by authors.
Introduction
- Write a sentence in introduction to show the economical loses (approximately) from Ph. nicotianae in tobacco.
- Write one or two sentences about Ph. nicotianae races (how many, importance).
- However, this disease is difficult to control via chemical means. Mention why?
- “Genome analysis showed that the Ph. nicotianae race 0 and race 1 contain expanded ATP-binding cassette transporter gene family” Write something to show the importance of containing that transporter, the importance of the finding after this genome analysis.
M&M
- “Genomic DNA was isolated using a standard phenol/chloroform protocol [29].” Describe briefly
- 2.1. “Ph. nicotianae (JM01) was isolated from infected tobacco plants…” which tobacco plant species, which variety if you know. Is it resistant or susceptible to Ph. nicotianae?
- What nanodrop did you use? Mention
- 2.8 Py. Infestans to be Ph. Infestans
- 2.12. Check if you have to write “multicollinearity”; there are 4 strains
Results
- 3.2 with19.15% space
- Add footnotes at Tables
- Check if results should be written below introduction and M&M below discussion in “plants”
Author Response
Question: Write a sentence in introduction to show the economical loses (approximately) from Ph. nicotianae in tobacco.
Response:Thanks for your kindly suggestion. We have added some description in the introduction section.
Question Write one or two sentences about Ph. nicotianae races (how many, importance).
Response:Thanks for your suggestion. Previous studies have identified four physiological races of this pathogen (0, 1, 2, and 3) in major tobacco cultivation areas worldwide. Among these races, the most common strain, race 0 has been the predominant race in the flue-cured area of North Carolina since 1931. Race 1 is pathogenic on N. plumbaginifolia, but its fitness was not equivalent to race 0, and it can be induced under intense selection pressure of continuous breeding of race 0 resistant varieties. Races 2 and 3 have been reported in South Africa, Connecticut, respectively. Now, we have added these descriptions in the introduction section in the manuscript.
Question However, this disease is difficult to control via chemical means. Mention why?
Response:Thanks for your question. There are many reasons for the difficulty of chemical control methods. First, there are many subspecies with different pathogenicity; second, the onset period is relatively wide, which may occur at any time; and third, the long-term use of chemical control methods can cause resistance to this pathogen.
Question “Genome analysis showed that the Ph. nicotianae race 0 and race 1 contain expanded ATP-binding cassette transporter gene family” Write something to show the importance of containing that transporter, the importance of the finding after this genome analysis.
Response:Thanks for your suggestion. We have revised this section according to your suggestions.
Question “Genomic DNA was isolated using a standard phenol/chloroform protocol [29].” Describe briefly
Response:Thanks for your suggestion. We have added relevant description of this protocol like this “100 mg samples were quickly ground in liquid nitrogen and then added to a centrifuge tube containing 500 μl lysate. Then, the homogenous material was centrifuged at 12000 g for 5 min. Next, the supernatant was transferred to another tube, and add equal volume phenol-chloroformisoamyl alcohol (25:24:1). Subsequently, the mixture was slowly homogenized for 2 min and centrifuged at 12000 g for 5 min. Then, the nucleic acids were precipitated by addition of 1,000 ul of 75% ethanol twice. Nucleic acids were pelleted at 12000 g for 10 min. The pellet was dried and was resuspended with 30 μL of ultrapure sterile water and stored at -80 °C.
Question 2.1. “Ph. nicotianae (JM01) was isolated from infected tobacco plants…” which tobacco plant species, which variety if you know. Is it resistant or susceptible to Ph. nicotianae?
Response:Thanks for your question. It is not a resistant or susceptible strain. It is a normal strain that cultured in Yunnan Province. While, we found some of the strains were infected in Yunnan Province in 2016. Then, we chosen one strain to isolated Ph. nicotianae (JM01) and stored it in the microbiology laboratory of the Institute of Tobacco Research Institute of Chinese Academy of Agricultural Sciences.
Question What nanodrop did you use? Mention
Response:OK, we have added the model type of nanodrop used in our manuscript “DNA concentration was evaluated using a NanoPhotometer spectrophotometer (IMPLEN, CA, USA).” In the M&M section.
Question 2.8 Py. Infestans to be Ph. Infestans
Response:Thanks for your kindly reminder. We have revised it.
Question 2.12. Check if you have to write “multicollinearity”; there are 4 strains
Response:Thanks for your kindly suggestion. We have revised “collinearity” as “multicollinearity” in this subtitle.
Question:3.2 with19.15% space
Response:Thanks for your kindly suggestion. We have deleted the extra space.
Question: Add footnotes at Tables
Response:Thanks for your kindly suggestion. We have added footnotes at Tables.
Question:Check if results should be written below introduction and M&M below discussion in “plants”
Response:Thanks for your kindly suggestion. We have revised the order of the sections.

Reviewer 3 Report
The manuscript is well written and has novel information on the pathogenicity mechanism of Phytophthora nicotianae. This research has addressed updated genome information of Phytophthora nicotianae and the research is relevant to current demand of exploring pathogenicity mechanisms of pathogen. However, the introduction need to be improved to include information of different races of Ph. nicotianae. The manuscript is well written and it is adding new valuable information on pathogenicity genes of Ph. nicotianae. The conclusion is well described to summarize the main findings of the research.
Author Response
Question: Comments and Suggestions for Authors
The manuscript is well written and has novel information on the pathogenicity mechanism of Phytophthora nicotianae. This research has addressed updated genome information of Phytophthora nicotianae and the research is relevant to current demand of exploring pathogenicity mechanisms of pathogen. However, the introduction need to be improved to include information of different races of Ph. nicotianae. The manuscript is well written and it is adding new valuable information on pathogenicity genes of Ph. nicotianae. The conclusion is well described to summarize the main findings of the research.
Response:Thanks for your kindly suggestions. We have added the description of Ph. Nicotianae races in the introduction section according to your suggestions. “Previous studies have identified four physiological races of this pathogen (0, 1, 2, and 3) in major tobacco cultivation areas worldwide. Among these races, the most common strain, race 0 has been the predominant race in the flue-cured area of North Carolina since 1931. Race 1 is pathogenic on N. plumbaginifolia, but its fitness was not equivalent to race 0, and it can be induced under intense selection pressure of continuous breeding of race 0 resistant varieties. Races 2 and 3 have been reported in South Africa, Connecticut, respectively.”

Round 2
Reviewer 2 Report
Dear authors
You answered my question and revised the recommendations. I don’t have further comments. My recommendation is: accept in present form.